# Effects of Mulberry Leaf Fu Tea on the Intestines and Intestinal Flora of Goto-Kakizaki Type 2 Diabetic Rats

**DOI:** 10.3390/foods12214006

**Published:** 2023-11-02

**Authors:** Changwei Liu, Hongzhe Zeng, Ronggang Jiang, Kuofei Wang, Jian Ouyang, Shuai Wen, Liyuan Peng, Hao Xu, Jianan Huang, Zhonghua Liu

**Affiliations:** 1Key Laboratory of Tea Science of Ministry of Education, Hunan Agricultural University, Changsha 410128, China; 2National Research Center of Engineering and Technology for Utilization of Botanical Functional Ingredients, Changsha 410128, China; 3Co-Innovation Center of Education Ministry for Utilization of Botanical Functional Ingredients, Changsha 410128, China; 4Key Laboratory for Evaluation and Utilization of Gene Resources of Horticultural Crops, Ministry of Agriculture and Rural Affairs of China, Hunan Agricultural University, Changsha 410128, China

**Keywords:** bacterial flora disorder, Type 2 diabetes, *Helicobacter*, mulberry leaf fu tea, α-glucosidase and α-amylase activities

## Abstract

Type 2 diabetes mellitus is a disease caused by hyperglycemia, an imbalance in the intestinal flora and disruption of the endocrine system. At present, it is primarily controlled through drug treatment and an improved diet. Mulberry leaf and fu brick tea were considered to have excellent hypoglycemic effects. This study used mulberry leaves and fu brick tea as raw materials to develop a dietary regulator that can assist in the prevention and alleviation of diabetes. The experiment used the Goto-Kakizaki (GK) rat model to investigate the hypoglycemic effect of mulberry leaf fu tea (MFT) and its influence on the intestinal flora of diabetic rats through methods including ELISA, tissue section observation and 16S RNA microbial sequencing. The results showed that, compared with the GK group, the intervention of mulberry leaf fu tea significantly reduced the activities of α-glucosidase (*p* < 0.05) and α-amylase (*p* < 0.05) in the duodenum of GK diabetic rats. The height of the duodenal villi was significantly reduced (*p* < 0.001), leading to decreased intestinal sugar absorption. At the same time, MFT alleviates the imbalance of intestinal flora caused by high blood sugar, promotes the growth of beneficial bacteria (*Lactobacillus*, *Bifidobacterium*, etc.), and inhibits the reproduction of harmful bacteria (*Blautia*, *Klebsiella*, *Helicobacter*, *Alistipes*, etc.). MFT helps reduce the secretion of toxic substances (lipopolysaccharide, *p* < 0.001), decreases oxidative stress and inflammation, mitigates organ damage, and improves symptoms of diabetes. Finally, the random blood glucose value of GK rats dropped from 22.79 mmol/L to 14.06 mmol/L. In summary, mulberry leaf fu tea can lower sugar absorption in diabetic rats, reduce the body’s oxidative stress and inflammatory response, regulate intestinal flora, and reduce blood sugar levels in GK rats. It is hinted that mulberry leaf fu tea could be used as a functional drink to help prevent the occurrence of diabetes.

## 1. Introduction

Type 2 diabetes mellitus (T2DM), also known as adult-onset diabetes, is a complex metabolic disease characterized by elevated blood glucose and lipid abnormalities [1]. The condition involves multiple factors, including insulin resistance, gut dysbiosis, and interactions between genetic and environmental factors [2,3]. Currently, most patients require oral medications or injectable therapies to manage their blood glucose levels. In treatment, especially in traditional Chinese medicine, the role of intestinal flora plays a crucial role in the therapeutic effect as it is closely related to drug metabolism and the progression of disease [4]. Clinically, diabetes can manifest as symptoms such as polyuria, polydipsia, polyphagia, polydipsia, and weight loss. Prolonged hyperglycemia causes damage to various organs, starting with microvessels and nerves and gradually extending to large blood vessels and major organs. It also affects the heart, skin, nerves, blood vessels, liver, eyes, and other systems [5]. These complications related to diabetes include diabetic nephropathy, diabetic eye disease, diabetic foot ulcers, diabetic heart disease, etc. These complications have a significant impact on the patient’s health and quality of life, and in some cases, they can even result in death [6].

Current research suggests a link between diabetes and the gut microbiota. Studies have shown that bioactive compounds and natural products that modulate the gut microbiota can significantly influence the mechanisms of disease in diabetes. Gut microorganisms mainly include *Firmicutes* (*Lactobacillus*, *Enterococcus*, and *Clostridium*), *Bacteroidetes*, *Proteobacteria* (*Enterobacterium*), *Actinobacteria* (*Bifidobacterium*), *Fusobacteria*, *Verrucomicrobia*, etc. [7]. The gut microbiota of patients with T2DM differs from that of healthy individuals in terms of abundance, proportion, and functionality. Compared to healthy individuals, the gut ecosystem of patients with T2DM is disturbed, with a decrease in beneficial bacteria (*Lactobacillus*, *Clostridium*, and *Bifidobacterium*) and an increase in pathogenic bacteria (*Escherichia*, *Enterococcus*, and *Clostridium*) [8,9]. One study found that Bifidobacterium levels were significantly higher in healthy individuals, while the level of Lactobacillus was considerably higher in patients with T2DM [10]. In a hyperglycemic environment, levels of Enterococcus increase while levels of *Bifidobacterium* and other *Bifidobacteria* in general decrease, exacerbating the progression of diabetes [11]. Opportunistic pathogenic bacteria (*Bacteroides*, *E. coli*, and *Desulfovibrio*) are also significantly increased in the gut of individuals with diabetes [12]. Researchers found that *Bifidobacterium*, *Lactobacillus*, *Ackermannia*, and *Faecalis* were negatively correlated with T2DM, while *Fusobacterium*, *Ruminococcus*, and *Cyanobacteria* were positively correlated with T2DM [13]. In diabetic patients, there is an abundance of *Bifidobacterium* and *Lactobacillus* and the *Bifidobacterium/Enterobacterium* (B/E) ratio. On the other hand, *Enterococcus* and *Enterobacteriaceae* increased significantly [14].

Disruption of the gut microbiota can cause pathogenic Gram-negative bacteria to release large amounts of endotoxin (lipopolysaccharide, LPS) in the gut. LPS is then transferred to the intestinal tissue [15]. LPS can induce intestinal inflammation, leading to insulin resistance and the expansion of adipose tissue, thereby compromising the integrity of the intestinal epithelium [16]. Once the intestinal mucosal epithelial barrier is damaged, an imbalance in the gut microbiota can occur, resulting in bacterial translocation. This, in turn, further promotes the production of harmful substances and worsens diabetes [17].

The treatment of diabetes often requires a variety of drugs that may be associated with various side effects. Natural products and their functional ingredients have become effective adjuncts to glucose-lowering therapy due to their advantages in lowering blood glucose levels and alleviating diabetic complications [18]. Compared to drug treatments, natural products are safer, have fewer toxicities and side effects, and can be used as dietary supplements [19]. They have a positive impact on improving the condition of patients with diabetes, reducing the development of complications, and enhancing the quality of life of patients [20]. Natural products have a wide range of applications in the treatment and management of diabetes [21]. Some natural products and plant extracts, such as tea, mulberry leaves, bitter melon, flavonoids, etc. [22,23,24,25], help control blood glucose levels, reduce insulin resistance, promote improvements in insulin sensitivity, and maintain stable blood glucose levels [26].

A study found that tea can reduce the abundance of Gram-negative bacteria and pathogenic bacteria and reduce damage to the intestinal barrier. This, in turn, reduces the displacement of LPS and inhibits the occurrence of inflammation and insulin resistance in the body [24]. Tea polysaccharides have been shown to restore the diversity of the intestinal microbiota of diabetic mice. They also increase and restore the relative abundance of *Achnospira*, *Victivallis*, *Roseburia*, and *Fluviicola* and increase the content of metabolites such as amino acids, which helps in reducing blood glucose and blood lipids in type 2 diabetic mice [27]. Matcha can increase the levels of *Coriobacteriaceae*, *Lactobacillaceae*, *Prevotellaceae*, and *Bifidobacteriaceae* and decrease the levels of *Bacteroidaceae*, *Ruminococcaceae*, *Helicobacteraceae*, and *Enterobacteriaceae* in diabetic rats [28].

An epidemiological study of 1118 people showed that drinking tea may reduce the risk of T2DM [29]. A clinical trial involving 1000 individuals showed that people with diabetes who consumed tea over an extended period could reduce urinary sugar excretion, alleviate symptoms, and even experience recovery [30]. Several studies highlight the potential of dark tea to reduce the risk of diabetes [31,32,33], which could become a significant public health strategy. In addition, mulberry leaves contain polysaccharides, flavonoids, alkaloids, volatile oils, and other active ingredients that can regulate the immune system and the gluconeogenesis process in diabetic patients while improving obesity and insulin resistance [34,35]. This study focuses on the duodenal structure, enzyme activity, oxidative stress, and intestinal flora. The aim was to investigate the hypoglycemic effect of mulberry leaf tea on hyperglycemic rats and its impact on gut flora. To develop a beverage that can help lower blood glucose levels, prevent the onset of diabetes, and improve the quality of life for patients.

## 2. Materials and Methods

### 2.1. Chemicals and Reagents

Mulberry leaf fu tea (MFT Hunan Gaojiashan Tea Co., Ltd., Yiyang, China), blood glucose test strips, and a blood glucose meter were purchased from Roche (Shanghai, China); α-amylase, α-glucosidase, superoxide dismutase (SOD), glutathione commercial kits such as glutathione peroxidase (GSH-PX), glutathione (GSH), and malondialdehyde (MDA) were purchased from Nanjing Jiancheng Bioengineering Institute (Nanjing, China); commercial kits for lipopolysaccharide (LPS) were purchased from Wuhan Huamei Bioengineering Co., Ltd. (Wuhan, China). Feed was supplied by Jiangsu Nantong Trophy Feed Co., Ltd. (Nantong, China).

### 2.2. Preparation and Dosage Selection of Mulberry Leaf and Fu Tea Extract

Preparation method of MFT extract: boil 1 g of MFT in 20 mL of distilled water (at 90 °C) (tea/distilled water = 1 g/20 mL) for 2 h, boil to obtain the MFT extract, then filter and collect the supernatant. Finally, freeze-dry the collected supernatant to powder using vacuum freeze-drying. The MFT extract was used in the following experiments, converted on the basis of the human body, consuming 9 g of mulberry leaf fu tea per day, and calculated on the basis of the body surface area normalization method [36]. The equivalent the dose of MFT extract for rats = 9 g ÷ 60 kg (body weight) × 6.2 (the conversion factor) × 22.5% (the leaching rate is approximate) ≈ 210 mg/kg.

### 2.3. Analysis of Chemical Components of MFT

Moisture content was determined according to GB/T 8304-2013 [37]; water extract content was determined according to GB/T 8305-2013 [38]; tea polyphenol content was determined according to GB/T 8313-2018 [39]; free amino acid content was determined according to GB/T 8314-2013 [40]; the anthrone colorimetric method was used to determine the soluble sugar content. The system analysis method was used to determine theaflavins (TFs), thearubigins (TRs) and theabrownin (TB) contents. High-performance liquid chromatography (HPLC) was used to determine EGGC (gallatechin gallate), EGC (epigallocatechin), ECG (epicatechin gallate), EC (epicatechin), GCG (epigallocatechin gallate), DL-C (catechin), gallate content of acid, and 3 alkaloids (caffeine, theobromine, and theophylline) [41]. (The chemical structure shows Appendix A).

The HPLC method was adapted from Zhou et al. [41] as follows: The chromatographic column is ECOSIL C18 (4.6 × 150 mm, 5 μm C/N EC181546 S/N, and 4I7501-11). The mobile phase A is ultrapure water. B is N,N-dimethylformamide/methanol/glacial acetic acid = 39.5:2:1.5. The detection wavelength is 278 nm, the column temperature is 30°C, the flow rate is 1 mL/min, and the injection volume is 10 µL. Gradient elution: 0 min, 91% A, and 9% B; 10 min, 86% A, and 14% B; 15 min, 77% A, and 23% B; 27 min, 64% A, and 36% B; 31 min, 64% A, and 64% B; 32 min, 91% A, and 9% B; 32 min; and stop.

### 2.4. Detection of Compounds in MFT

The LC-MS method was based on that of Han et al. [42], with slight modifications. The details are as follows: LC-MS detection using the ThermoScientific OE120+ mass spectrometry platform and HPLC-QTrap-MS/MS for determination, which includes MRM quantification and ultra-efficient HPLC-Q-TOF target identification. HPLC conditions: Waters X-selectTMHSST3 column, 2.5 μm, 2.1 × 150 mm, with mobile phase A being 0.1% formic and mobile phase B being 0.1% formic acid–acetonitrile. After collecting the supernatant, the column temperature is set at 50 °C and 0.4 mL/min gradient elution: 0 min, 98% A, and 2% B; 2 min, 98% A, and 2% B; 15 min, 0% A, and 100% B; 17 min, 0% A, and 100% B; 17.1 min, 98% A, and 2% B; and 20 min, 98% A, and 2% B. Positive ion mode: curtain gas at 35 psi, collision gas medium, ionization voltage at 5500 V, temperature at 550 °C, spray gas at 60 psi, and auxiliary heating gas 60 psi. Raw LC-MS data were imported into Compound Discoverer 3.0 for baseline filtering, peak identification, integration, retention time correction, and peak alignment. This process resulted in the production of retention time, mass-to-charge ratio, and peak intensity data matrices. The 80% rule was used to remove missing values, retain non-zero variables present in at least 80% of the samples, and impute missing values. Finally, to reduce errors due to sample preparation and instrument instability, the data summation was normalized.

### 2.5. Animal Experiments

A total of 20 SPF male Goto-Kakizaki (GK) rats and 8 Wistar rats aged 10–12 weeks were purchased from Jiangsu Changzhou Cavins Experimental Animal Co, Ltd. (Changzhou, China), Experimental Animal Production Licence No: SCXK (Su) 2021-0013. Animal experiments were approved by the Animal Experimentation Committee of Hunan Agricultural University [Approval Number: Lun Shen Ke 2021 No. (2137)]. The temperature of the animal room was maintained at 24–26 °C, the light on time/the light of time was 12/12 h cycle, and there was free access to water and food. Food, distilled water, and bedding were changed once a day to keep the cage clean and hygienic. After all rats had been adaptively fed and adopted the diet for one week, the random blood glucose levels of the rats in each group were measured. The successfully modeled GK rats were randomly divided into two groups: the hyperglycemia model group (GK) and the MFT group (T). In addition, 8 Wistar rats comprised the normal control group (NC). Wistar rats were fed standard chow (energy was 3.6 kcal/g, composed of 19.4% protein, 70.6% carbohydrate, and 10% fat). GK rats were fed a high-fat diet (energy: 5.1 kcal/g, consisting of 19.4% protein, 20.6% carbohydrate, and 60% fat). MFT was used for continuous intervention for 4 weeks, and random blood glucose levels were measured twice a week. At the end of the experiment, the rats fasted for 12 h and were then anesthetized via intraperitoneal injection of sodium pentobarbital (30 mg/kg). Blood was collected from the aorta, stored in a refrigerator at 4 °C for 2 h and centrifuged at 3500 rpm in a high-speed centrifuge at 4 °C for 10 min. The top serum was aspirated, aliquoted into EP tubes, and frozen at −80 °C for later use [43]. The visceral tissues of the rats were sequentially removed, soaked in 10% formalin, and fixed upside down for histopathological examination.

### 2.6. Determination of Biochemical Indicator

The activities of α-amylase and α-glucosidase in the duodenum of rats in each group and the lipopolysaccharide (LPS), superoxide dismutase (SOD), glutathione peroxidase (GSH-PX) in the serum, GSH, malondialdehyde (MDA) content were determined using kit methods.

### 2.7. Rat Tissue Morphological Observation

The duodenum, kidney, and other tissues were fixed in 4% paraformaldehyde for 24 h. Subsequently, the tissue was dehydrated, paraffin-embedded, and sectioned. The tissue sections were then stained with Hematoxylin-eosin (H&E). The morphological changes of the duodenum and kidneys in each group were observed under a microscope and photographed for documentation.

### 2.8. 16S RNA Analysis of Rat Intestinal Flora

The intestinal contents were resuspended in sterile water, and bacterial DNA was extracted using a commercial kit (QiAamp DNA Stool Kit, Berlin, Germany) following the kit instructions, followed by agarose gel electrophoresis, and Nanodrops were used to determine the purity and concentration of the DNA. Primers F341 (5′-CTAYGGGRBGCASCAG-3′) and R806 (5′-GGACTACNNGGGTATCTAAT-3′) were used to amplify the V3–V4 domain of the bacterial 16S rRNA gene. Mix the PCR products from the previous step in a 1:1 mass ratio, as determined using the electrophoresis quantification results. After mixing, use the OMEGA DNA Purification Column for column purification. After electrophoresis on a 1.8% agarose gel at 120 V for 40 min, the target fragment is excised and recovered.

Purified libraries were sequenced using the Illumina PE250bp (Illumina, San Diego, CA, USA). After quality filtering, the raw reads were clustered into operational taxonomic units (OTUs) using UPARSE. The species sequence alignments were then annotated using the Ribosomal Database Project (RDP). A rare analysis of the Shannon index of the gut microbiota was performed using Mothur. R packages were used to perform Principal Coordinate Analysis (PCoA) and Hierarchical Cluster Analysis [44].

### 2.9. Data Statistics

All experiments were performed with three biological replicates, and the data are expressed as the mean ± standard deviation. SPSS 26.0 software was used to conduct a one-way analysis of variance and an LSD test to analyze the data differences between different groups. Graphs were drawn using GraphPad Prism 9.5. There is a statistically significant difference when *p* < 0.05, * *p* < 0.05, ** *p* < 0.01, *** *p* < 0.001, and **** *p* < 0.0001.

## 3. Results

### 3.1. Chemical Composition of MFT

The major constituents of MFT extract are shown in Figure 1. In this study, the catechins, alkaloids, and gallic acid in MFT were tested. The results are shown in Figure 1. The content of catechins in MFT accounts for 1.12% of its dry weight; soluble sugar is 5.86%; theaflavins are 0.13%; thearubigins are 2.07%; theabrownins are 14.91%; free amino acids are 2.49%, and polyphenols is 9.98%. Compared to green tea, MFT has lower t levels of catechins, alkaloids, and other substances but higher levels of tea pigments. After ordering fermentation, blending, flowering, and other processes, the irritating substances in MFT were oxidized and polymerized into different compounds, resulting in a milder and less irritating taste.

Figure 1B shows the particle flow diagram of MFT using LC-MS. Through LC-MS detection and analysis, they identified a wide range of compounds in the tea, amounting to a total of 376 species (Appendix A). These compounds fall into 11 main classes: catechins and their derivatives, alkaloids, flavonoids, flavonoid glycosides, phenolic acids and their products, amino acids and their derivatives, organic acids, fatty acids, sugars and glycoside classes, purine bases, and other unclassified substances. According to peak area analysis, the main compounds found in MFT include caffeine, catechins, sugars, amino acids, and other substances. These ingredients are the main components of fu tea and contribute to its distinctive flavor and properties. In addition, the inclusion of mulberry leaves and leaves in the mixture resulted in the detection of some unique compounds through LC-MS analysis. These compounds include myricetin 3-O-β-D-galactopyranoside, naringin, astragalus, hesperetin, new orange dermatitis, etc. It should be noted that the concentrations of these compounds are relatively low. In general, the main constituents of MFT are still substances commonly found in fu brick tea. However, the addition of mulberry leaves brings some additional compounds to the tea. These compounds affect the taste and quality of the tea. It is the diverse chemical composition that forms the chemical basis of MFT and determines its health benefits. The presence and interaction of these compounds give MFT several beneficial health properties.

### 3.2. Effects of MFT Extract on Blood Glucose and Related Biochemical Indicators in Diabetic Rats

T2DM is a systemic metabolic and endocrine disease characterized by the dysregulation of glucose and lipid metabolism. This study used the spontaneous diabetic GK rat model to investigate the effect of MFT extract on blood glucose levels in diabetic rats. The results are shown in Figure 2. After 1 week of adaptive feeding under high-fat diet conditions, the blood glucose levels of GK rats exceeded 20 mmol/L, indicating that the GK rat T2DM model was successfully established. We randomly measured the blood glucose level of rats in each group every week and found that the blood glucose level of rats in the GK group was significantly higher than that in the NC group. This trend remained constant throughout the experimental period (*p* < 0.0001). However, the blood glucose levels of rats in the T group receiving MFT extract were significantly reduced (*p* < 0.001) and reached their lowest level in the fourth week. These results indicate that MFT extract can effectively reduce blood glucose levels in GK rats.

A-glucosidase and α-amylase are critical enzymes for carbohydrate digestion in the body and are essential for the breakdown and metabolism of carbohydrates. Their activity is closely related to postprandial blood glucose levels in individuals with diabetes. We monitored the levels of α-glucosidase and α-amylase in the duodenum of rats in each group. The results showed that the levels of α-glucosidase and α-amylase in the duodenum of rats in the GK group were higher than those in the NC group. However, in the T group of rats given MFT extract, the levels of these two enzymes were significantly reduced, thereby decreasing the intestinal absorption of sugar.

### 3.3. Effects of MFT Extract on Biochemical Indicators in Diabetic Rats

In a state of insulin resistance, pancreatic beta cells continue to produce excessive insulin, resulting in oxidative stress. This intense oxidative stress is not only associated with impaired glucose metabolism, insulin resistance, and the development of diabetes, but it also promotes the further development of complications related to diabetes. To investigate the effect of MFT extract on the antioxidant status of diabetic rats, we measured the activities of SOD and GSH-PX and the levels of MDA in the serum. As shown in Figure 3, compared with the NC group, the serum MDA level of the hyperglycemic group had no significant change (*p* < 0.05). However, the intervention of MFT significantly reduced the serum MDA level of the hyperglycemic rats (*p* < 0.0001), even falling below the level of the NC group (*p* < 0.0001). GSH-Px is an important peroxide-decomposing enzyme that is widely distributed in the body. Serum GSH-Px levels were often elevated in diabetic patients. As can be seen from Figure 3, the GSH-Px level in the serum of the hyperglycemic group was significantly increased (*p* < 0.001) compared to the NC group, whereas the MFT intervention significantly decreased the GSH-Px level in the serum of the hyperglycemic rats (*p* < 0.0001) and reduced it to the level of the NC group (*p* < 0.05). As shown in Figure 3, MFT intervention can increase the levels of SOD (*p* < 0.05) and GSH (*p* < 0.05) in the serum of hyperglycemic rats. These results show that MFT intervention can significantly improve systemic oxidative stress damage in hyperglycemic rats, thereby reducing the incidence of diabetic complications.

LPS is a biomolecule that can trigger the body’s inflammatory response, and its role is to release inflammatory mediators, ultimately leading to insulin resistance [45]. We measured the levels of the pro-inflammatory substance LPS and the inflammatory mediator Il-1β in the serum. The results of the study showed that in the GK group (the diabetes model group), serum levels of LPS and Il-1β were significantly increased compared to the NC group. However, in the T-group rats that received the MFT extract, we observed a significant decrease in serum levels of LPS and Il-1β. It suggests that MFT extract may have the potential to inhibit the inflammatory response induced by endotoxin LPS, thereby helping to reduce the incidence of insulin resistance.

### 3.4. Effects of MFT Extract on Duodenal and Kidney Morphology in Diabetic Rats

High blood glucose is one of the main symptoms of diabetes, and long-term, uncontrolled high blood glucose can cause damage to the kidneys. To further analyze the effect of dark tea on the kidney structure of diabetic rats, we used H&E staining to stain these two organs. The results are shown in Figure 4: In the NC group, the renal cortical tubules were closely arranged, and the glomerular tissue structure was clear, without any abnormality. In the rats in the GK group, protein casts appeared in the renal tubules, plasma proteins leaked from the renal capsules, the renal tubular epithelial cells expanded in a vacuole-like manner, the renal tubular structure significantly expanded, and the glomerular volume increased. However, in the T-group rats, the morphology of the kidney tissue was similar to that of the NC-group rats, with no abnormal findings, and significantly different from that of the GK group.

The duodenum is the leading site of digestion and absorption in the gastrointestinal tract (Figure 4). H&E sections of the duodenum showed that the height of the intestinal villi in GK rats was significantly higher than in the NC and T groups. Still, the density of the intestinal villi did not change significantly. It hints that diabetes may lead to longer duodenal villi, which increase the absorptive surface area of the intestinal mucosa. This could be a compensatory mechanism for the body’s impaired glucose uptake and utilization and negative nitrogen balance. However, after oral administration of MFT extract to GK rats, there was a significant reduction in the height of the duodenal villi. This reduction resulted in a decrease in the absorptive surface area of the intestinal mucosa. Surface area of the intestinal mucosa, thereby reducing the body’s absorption of sugar.

### 3.5. Effect of MFT on Intestinal Flora of Hyperglycemic Rats

Diabetes is closely associated with intestinal microorganisms. We sequenced the V3–V4 hypervariable region of 16S rRNA from the gut flora of rats treated for 4 weeks and analyzed the effect of MFT extract on the gut flora of diabetic rats. As shown in Figure 5D, the number of OUTs in each group detected via 16S RNA sequencing is as follows: 621 in the NC group, 287 in the GK group, and 388 in the T group. The Simpson index of gut flora calculated from the OTU results was commonly used to assess the richness and diversity of the microorganisms in the rat gut. As shown in the figure, the ACE and Simpson indices of each test group were significantly different. Compared to the NC group, the Ace and Shannon index of the bacterial flora were considerably reduced in the GK group. This indicates that diabetes may lead to a reduction in the diversity of the gut flora in rats, resulting in gut dysbiosis. In addition, OTU-based Principal Coordinates Analysis (PCoA) and non-metric Multidimensional Scaling (NMDS) analysis were used to compare the overall effect of tea extract on the gut flora of diabetic rats. As shown in Figure 5D, the gut flora structures of the NC, GK, and T groups exhibited significant differences, suggesting notable variations in species composition among the groups. Among them, the GK group and the NC group showed substantial separation.

Additionally, with the intervention of MFT extract, the GK group and the T group exhibited significant division in PCOA and stump clustering. These findings indicate that the intestinal flora of diabetic rats was significantly improved after the intervention with MFT extract. With the intervention, the diversity and richness of the intestinal microbiota in diabetic rats were partially restored.

At the phylum level (Figure 6A), the gut microbiota is typically composed of a diverse range of bacterial species. Among these, *Bacteroidetes*, *Firmicutes*, *Verrucomicrobia*, *Proteobacteria*, and *Actinobacteria* are commonly dominant bacterial phyla. However, in the diabetic GK group, the abundance of *Firmicutes*, *Bacteroidetes*, and *Verrucomicrobia* was higher than that of the NC group. On the contrary, the abundance of *Proteobacteria* and *Cyanobacteria* was significantly higher in the GK group than in the NC group, leading to alterations in the Firmicutes/Bacteroidetes ratio. However, after oral administration of MFT to the GK group, there was no significant increase in the Firmicutes. But the *Firmicutes/Bacteroidetes* ratio showed a tendency similar to that of the NC group. This suggests that MFT may help restore the balance of the gut microbiota in GK diabetic rats and inhibit the proliferation of cyanobacteria. In addition, significant changes were observed in certain bacterial groups at the family level following diabetes and MFT treatments.

Families such as *Lachnospiraceae*, *Ruminococcaceae*, *Erysipelotrichaceae*, *Bacteroidaceae*, *Eubacteriaceae*, *Lactobacillaceae*, *Porphyromonadaceae*, and *Bifidobacteriaceae* dominate the gut microbiota. Diabetes may lead to a significant increase in the abundance of some bacterial families, including *Lachnospiraceae*, *Erysipelotrichaceae*, *Eubacteriaceae*, *Bifidobacteriaceae*, and *Helicobacteraceae*. Meanwhile, some other family levels of bacteria may be reduced in diabetes *(ruminococcus-ceae*, *Micrococcaceae*, and *Verrucomi-crobiaceae*). However, after the administration of MFT extract, the abundance of these bacterial families returned to levels close to the NC group (Figure 6B).

At the genus level, *Clostridium-XlVa*, *Blautia*, *Bacteroides*, *Ruminococcus*, *Eubacterium*, *Pseudoflavonifractor*, *Faecalicoccus*, *Flavonifractor*, *Lactobacillus*, *Akkermansia*, and other bacterial genera are dominant groups in the rat intestine (Figure 6C). Compared to the NC group, the GK group showed enrichment of *Blautia*, *Klebsiella*, *Roseburia*, *Eubacterium*, *Helicobacter*, and *Alistipes*, while the abundance of *Clostridium-XlVa* and *Ruminococcus* was reduced. *Flavonifractor*, *Lactobacillus*, and *Oscillibacter* genera were reduced. However, the administration of MFT extract appeared to reverse this trend. In addition, *Akkermansia* is considered to be a beneficial bacterial genus, whereas *Desulfovibrionaceae* is deemed to be a potentially harmful bacterium. As shown in Figure 6, diabetes may promote the proliferation of bacterial groups, such as *Desulfovibrionaceae*, while inhibiting the growth of bacterial groups, such as *Akkermansia*. However, after oral administration of MFT extract, the abundance of bacterial genera such as Desulfovibrionaceae and *Akkermansia* did not appear to return to normal levels, which may mean that MFT extract has a limited effect on these bacterial genera.

The gut microbiota is a highly complex ecosystem in which different types of microorganisms influence and interact with each other. To gain a deeper understanding of the interaction between dominant bacterial genera in patients with diabetes, we performed Spearman correlation analysis to examine the relationship between the dominant bacterial genera in each group. The analysis results showed (Figure 6D) that probiotic *Lactobacillus.apodemi* was positively correlated with *Akkermansia.muciniphila*, *Ru-minococcus.bromii*, *Blautia.glucerasea*, *Clostridium.disporicum*, and other bacterial genera. This indicates a positive mutual promoting relationship between them. There is a negative correlation between *Pseudoflavonifractor.capillosus*, *Clostridium.glycyrrhizinilyticum*, *Flavonifractor.plautii*, *Bacteroides.sartorii*, and other bacterial genera, suggesting mutual inhibition among them. In addition, *Helicobacter.ganmani* was negatively correlated with *Pseudoflavonifractor.capillosus*, *Eubacterium.coprostanoligenes*, *Bacteroides.sartorii*, Clostridium. Aldenense, and other bacterial genera, hinting that these genera may inhibit the proliferation of *Helicobacter.ganmani*. These findings provide a deeper understanding of the impact of interactions between different bacterial genera on the gut microbiota.

Further analysis of the changes in bacterial abundance at the genus level (Figure 7) showed that diabetes caused a significant increase in the abundance of bacteria such as *Alistipes*, *Roseburia*, *Lactococcus*, *Peptococcus*, and *Helicobacter* in the intestinal tract of rats, as well as bacteria such as *Lactobacillus*, *Flavobacterium*, *Oscillibacter*, and *Murimonas*. The abundance was significantly reduced. However, after the GK rats consumed MFT extract, the relative abundance of *Alistipes*, *Roseburia*, *Lactococcus*, *Peptococcus*, *Helicobacter*, and other bacteria decreased. In contrast, the proportion of *Lactobacillus*, *Flavobacterium*, *Oscillibacter*, *Murimonas*, and other bacteria contributing to this flora returned to normal levels. This suggests that MFT extract may help maintain the balance of intestinal flora in diabetic rats.

The Linear Discriminant Analysis Effect Size (LEfSe) value indicates the degree of contribution of the microorganism to differentiating between groups. The larger the LDA value, the more significant the role of the microorganism in the differences between groups. We set an LDA threshold of 3 to identify species with significant differences in microbiota abundance. The results are shown in Figure 8. The NC group mainly enriched 28 dominant bacterial genera, including *Flavobacteriales*, *Vibrionaceae*, *Bilophila*, *Vibrio*, *Sutterella*, *Algoriphagus*, *Pseudomonadaceae*, *Clostridium-IV*, *Acetatifactor*, *Ruminococcus*, etc. Group T mainly enriched 20 dominant bacterial genera, including *Desulfovibrio*, *Eisenbergiella*, *En-terorhabdus*, *Proteobacteria*, *Allobaculum*, *Oscillibacter*, *Pseudoalteromonas*, *Alteromonadales*, *Clostridiaceae*, *Romboutsia*, etc. The GK group mainly enriches 17 dominant bacterial genera, including *Roseburia*, *Ruminococcus*, *Erysipelotrichia*, *Coprococcus*, *Holdemania*, *Helicobacter*, *Campylobacterales*, *Helicobacteraceae*, *Peptococcus*, *Rikenellaceae*, *Alistipes*, *Lactococcus*, etc. It hinted that *Roseburia*, *Ruminococcus*, *Helicobacter*, *Helicobacteraceae*, *Peptococcus*, *Rikenel-laceae*, *Alistipes*, and other bacteria constitute the core flora in GK diabetic rats.

### 3.6. Analysis of the Correlation between Intestinal Bacteria and Physiological Indicators of the Body

To further explore the potential role of different bacterial genera in diabetes, the Spearman correlation analysis was used to examine the correlation between other bacterial genera between the GK group and the T group, as well as the physiological indicators of mice (Figure 9). The results showed that the bacterial genera associated with the inflammatory factor IL-1β included Helicobacter, *Alistipes*, *Peptococcus*, *Roseburia*, *Clostridium-XVIII*, etc. These bacteria may be involved in the inflammatory response. *Flavobacterium*, *Ruminococcus*, *Os-cillibacter*, *Phascolarctobacterium*, and other bacterial genera were negatively correlated with IL-1β. These bacteria may play a role in the inflammatory response. *Peptococcus*, *Clostridium-XlVb*, *Roseburia*, and *Faecalicoccus* are positively associated with MDA levels, and these bacteria may be related to oxidative stress. LPS was negatively correlated with *Oscillibacter* and *Flavobacterium*. In contrast, LPS was positively correlated with duodenal villus height and glomerular area, suggesting that LPS may be related to the structure of the duodenum and kidneys. α-Glucosidase is negatively correlated with *Lactobacillus* and *Clostridium-XlVb*, and these bacteria may be associated with sugar metabolism. Duodenal villus height is positively correlated with *Peptococcus*, *Alistipes*, *Desulfovibrio*, *Roseburia*, *Clostridium-XlVb*, and *Faecalicoccus.* This may indicate that these bacteria are related to the size of the villi in the small intestine. The levels of MDA and IL-1β were positively correlated with duodenal villus height and glomerular area, indicating a potential relationship between MDA and IL-1β and alterations in the villus and glomerular area of the small intestine. These results suggest that in diabetic mouse models, there is an enrichment of bacterial flora associated with glucose and lipid metabolism, barrier dysfunction, inflammatory response, and oxidative stress.

## 4. Discussion

T2DM is a prevalent chronic metabolic disease characterized by high blood glucose (hyperglycemia) and insulin resistance. Treatment approaches for T2DM currently include lifestyle interventions and pharmacotherapy, with a reduction in carbohydrate and sugar intake playing a pivotal role in mitigating disease progression [46]. In particular, regulating postprandial blood glucose levels has been shown to be an effective strategy for treating T2DM and preventing its complications [47]. A hyperglycemic state can affect the absorption and metabolism of sugars in the duodenum, where alpha-amylase and alpha-glucosidase are essential enzymes. They can promote the breakdown of starch, sucrose, maltose, and other carbohydrates in the diet into simple sugars. These simple sugars are then absorbed by the epithelial cells of the upper small intestine and enter the bloodstream, causing a rise in blood glucose [48]. Therefore, α-amylase and α-glucosidase have critical clinical applications in the treatment and control of diabetes. This study found that the levels of alpha-amylase and alpha-glucosidase in the duodenum increased under hyperglycemic conditions compared with the NC group. However, after oral administration of MFT extract, the levels of α-amylase and α-glucosidase in diabetic rats were significantly reduced, thereby reducing the ability of diabetic rats to absorb sugar. This finding suggests that MFT extract may have the potential to inhibit α-glucosidase activity and help reduce postprandial blood glucose elevations (Figure 2).

Oxidative stress plays a significant role in the pathogenesis of diabetes. Under conditions of high blood sugar, the activity of antioxidant enzymes in the body, such as SOD, CAT, GSH, etc., decreases, resulting in damage to the antioxidant system. The body’s ability to scavenge free radicals is reduced [49]. In particular, the levels and activity of the antioxidant enzymes SOD, CAT, and GSH in pancreatic islet B cells are low. This causes excessive ROS to accumulate in the islets, directly damaging the islet cells and affecting insulin synthesis and secretion signaling pathways. Indirectly, it also indirectly damages pancreatic islet B cells, thereby promoting the onset of diabetes [50]. It is worth noting that relevant studies have found that the activity of GSH-PX in the body may be reduced when diabetes occurs [51]. MFT intervention significantly reduced serum MDA levels in hyperglycemic rats (*p* < 0.0001), below the level of the NC group (*p* < 0.0001). In addition, MFT intervention can increase the levels of SOD (*p* < 0.05) and GSH (*p* > 0.05) in the serum of hyperglycemic rats (Figure 3). These results show that MFT intervention can significantly improve systemic oxidative stress damage in hyperglycemic rats, thereby reducing the incidence of diabetic complications. However, we found that serum GSH-PX activity was substantially higher in GK diabetic rats than in the NC group. This difference may be due to the different T2DM models. The activity of GSH-PX in the body is not as high as it could be, but a proper balance must be maintained. GSH-PX activity in the serum of GK rats was significantly reduced after oral administration of MFT extract. That helped to correct the abnormal GSH-PX enzyme activity caused by hyperglycemia. It suggests that MFT may have potential benefits for alleviating oxidative stress conditions in patients with diabetes.

Inflammatory processes play a vital role in the development of T2DM, including multiple pathological mechanisms such as glucotoxicity, lipotoxicity, oxidative stress, and endoplasmic reticulum stress [52]. The intestine, as a primary site of digestion and absorption, is also one of the most essential organs of the host immune system. Studies have shown that patients with T2DM often have low-grade inflammation. Intestinal inflammation leads to changes in the structure of the intestines. Poisonous substances (such as LPS, etc.) can enter the blood and exacerbate the inflammatory response structure of the intestines [53]. LPS and IL-1β levels in the serum of rats in the GK group were significantly higher than those in the NC group. This is because a hyperglycemic environment causes intense oxidative stress in the intestine, which disrupts the intestinal barrier. This disruption allows harmful substances, such as LPS, to enter the blood, triggering an inflammatory response in the intestine and thus promoting the secretion of inflammatory factors such as IL-1β. Subsequently, factors such as LPS and IL-1β circulate throughout the body, further exacerbating the symptoms of diabetes. We found that MFT extract can reduce the levels of LPS and IL-1β in the blood and reduce the body’s inflammatory response. MFT may have the potential to regulate inflammatory processes in the gut and help improve the condition of T2DM patients (Figure 3).

Diabetes is a metabolic disease often associated with hyperglycemia (high blood sugar). The duodenum is the central part of the body where sugar is digested and absorbed. Hyperglycemia can lead to changes in the structure and function of the duodenal villi. These villi are tiny projections responsible for absorbing and transporting nutrients, and changes in these villi can affect how efficiently sugar and other nutrients are absorbed [54,55]. According to the observation of H&E sections of the duodenum, the height of the intestinal villi in GK rats was significantly higher than in the NC and T groups. Still, the density of the intestinal villi did not change much (Figure 4). Diabetes may cause the duodenal villi to lengthen, thereby increasing the absorptive surface area of the intestinal mucosa, which may be a compensatory mechanism for the body’s impaired glucose uptake and utilization and negative nitrogen balance. However, after the oral administration of MFT extract to GK rats, this reduction led to a decrease in the absorptive surface area of the intestinal mucosa, resulting in reduced absorption of sugar by the body.

Hyperglycemia increases the filtration load on the glomerulus, resulting in an increase in the glomerular filtration rate and glomerular swelling [56]. In addition, high blood glucose can damage renal tubular cells and interfere with their normal p-glucose and protein processing, resulting in diabetes and proteinuria [57]. In the GK group, we observed protein casts in the renal tubules, plasma protein leakage into the renal capsule, vacuolar expansion of the renal tubules in rats, significant development of the renal tubular structure, and an increase in glomerular volume. However, after MFT, the morphology of the kidney tissue was similar to that of the rats in the NC group, and no abnormalities were found, which was significantly different from that of the rats in the GK group.

The gut is the most significant internal environment in the human body. It is prosperous in terms of the number and types of microorganisms and is closely linked to human health. Recent studies have shown that an imbalance in gut microbial communities is nearly related to the onset and development of T2DM. Intestinal microorganisms influence the body’s metabolism and immune function by participating in various physiological processes such as glucose metabolism, lipid metabolism, and immune regulation. However, after MFT, the morphology of the kidney tissue was similar to that of the rats in the NC group, and no abnormalities were found. The situation was obviously different from that of rats in the GK group [58]. There is mounting evidence that changes in the structure of the gut microbiota may be associated with diabetes and its complications, such as diabetic neuropathy and cardiovascular disease [15]. Studies have shown that the proportions of *Firmicutes* and *Clostridia* were significantly reduced in patients with diabetes. Furthermore, the reduction in Bacteroidetes and Proteobacteria, along with the increase in the ratio of *Betaproteobacteria*, was positively correlated with plasma glucose levels [59]. These changes may lead to intestinal metabolic disorders and trigger immune and inflammatory responses that are associated with the pathogenesis of diabetes and its complications.

Hyperglycemia can lead to reduced diversity of gut microbiota and dysbiosis in rats [60]. However, research results of MFT extract showed that it significantly improved the disorder of the intestinal microbiota in diabetic rats and restored the diversity and richness of the intestinal microbiota. Through analysis at different classification levels, we found that at the phylum level, compared with the NC group, the number of Firmicutes increased and the number of Proteobacteria decreased in the GK group, resulting in a decrease in the Firmicutes/Bacteroidetes ratio. However, after oral administration of MFT extract, the abundance of Firmicutes in the gut microbiota of GK rats increased and the abundance of harmful Bacteroidetes decreased, further increasing the Firmicutes/Bacteroidetes ratio (Figure 5).

Bacteroidetes are Gram-negative bacteria whose cell walls consist mainly of LPS. Following the death of Gram-negative bacteria, the degradation of the cell wall can release LPS, triggering an inflammatory response that leads to an immunostimulatory cascade. At the genus level, hyperglycemia enriched genera such as *Blautia*, *Klebsiella*, *Roseburia*, *Eubacterium*, *Helicobacter*, and *Alistipes* while reducing the abundance of genera such as Clostridium-XlVa, *Ruminococcus*, *Flavonifractor*, *Lactobacillus*, and *Oscillibacter.* However, MFT extract seems to reverse this trend. It should be noted that *Akkermansia* was considered a beneficial genus of bacteria, whereas *Desulfovibri-onaceae* is a potentially harmful bacteria. Research shows that diabetes can promote the growth of bacterial groups such as *Desulfovibrionaceae*, while inhibiting the growth of bacterial groups such as *Akkermansia.* However, after administering the MFT extract, the abundance of *Desulfovibrionaceae* and *Akkermansia* did not seem to return to normal levels, which may mean that the MFT extract has a limited effect on these bacterial genera and cannot regulate the abundance of these two genera (Figure 6).

Many studies have shown that harmful bacteria in the intestines of diabetic patients, including *Escherichia coli*, *Staphylococcus*, *Proteus*, *L.welshimer*, *Veillonella*, *Clostridium Prazmowski*, *Streptococcus*, *Peptostreptococcus*, *Fusobacterium*, *Clostridium*, *Klebsiella*, *Prevotella*, *Clostridium tetani*, *Bacteroides*, *Veillonella*, *Atopobium*, *Rumenococcus*, *Bacteroides*, *Desulfovibrio*, *Monilia albican*, etc. were significantly enriched [22,23,61]. These predominant harmful bacteria dominate the flora of diabetic patients. Using LDA analysis, we determined the characteristic bacterial flora of both the GK and T groups. The GK group mainly included 17 dominant bacterial genera, such as *Roseburia*, *Ruminococcus*, *Erysipelotrichia*, *Coprococcus*, *Holdemania*, *Helicobacter*, *Campylobacterales*, *Helicobacteraceae*, *Peptococcus*, *Rikenellaceae*, *Alistipes*, *Lactococcus*, etc. The T group enriched 20 major dominant bacterial genera, including *Enterorhabdus*, *Proteobacteria*, *Allobaculum*, *Oscillibacter*, *Pseudoalteromonas*, *Alteromonadales*, *Clostridiaceae*, *Romboutsia*, etc. However, intervention with MFT extract seemed to change the relative abundance of some bacterial groups. For example, the relative abundance of bacteria such as *Lactobacillus*, *Flavobacterium*, *Oscillibacter*, and *Murimonas* increased, while the relative abundance of *Listipes*, *Roseburia*, *Lactococcus*, *Peptococcus*, *Helicobacter*, and other bacterial groups decreased (Figure 8). Meanwhile, serum LPS was positively correlated with duodenal villus height, and glomerular area. IL-1β was positively correlated with *Helicobacter*, *Alistipes*, *Peptococcus*, *Roseburia*, and *Clostridium-XVIII* (Figure 9). It shows that these bacteria can produce harmful substances and exacerbate the body’s inflammatory response.

These results suggest that MFT extract may have a beneficial effect on the gut microbiota of diabetic patients by adjusting the relative abundance of specific groups of bacteria and helping to improve the balance of gut microbes. However, further research is needed to gain a deeper understanding of the impact of these microbial changes on diabetes and related metabolic diseases, as well as the mechanism of action of MFT in treatment.

## 5. Conclusions

In conclusion, drinking MFT could have a number of beneficial effects on GK diabetic rats. First, MFT reduced the activity of α-glucosidase and α-amylase in the duodenum, inhibited the elongation of duodenal villi, and reduced sugar absorption. At the same time, the intervention of MFT improved the intestinal microbiota disruption caused by hyperglycemia and increased the diversity and stability of microorganisms. It promotes the growth of beneficial bacteria (*Lactobacillus*, *Flavobacterium*, and *Oscillibacter*) while inhibiting the proliferation of harmful bacteria (*Blautia*, *Klebsiella*, *Roseburia*, *Eubacterium*, *Helicobacter*, and *Alistipes*), helping to reduce the secretion of toxic substances. MFT helps to reduce oxidative stress and inflammation in the body, minimize organ damage, improve insulin resistance, and ultimately achieve the goal of lowering blood sugar.

## Figures and Tables

**Figure 1 foods-12-04006-f001:**
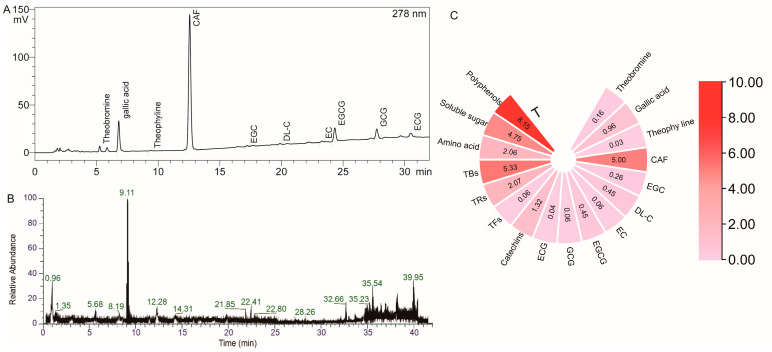
Ingredient map of MFT and content of each substance. (**A**): HPLC spectrum. (**B**): LC-MS particle flow diagram. (**C**): MFT conventional chemical composition content.

**Figure 2 foods-12-04006-f002:**
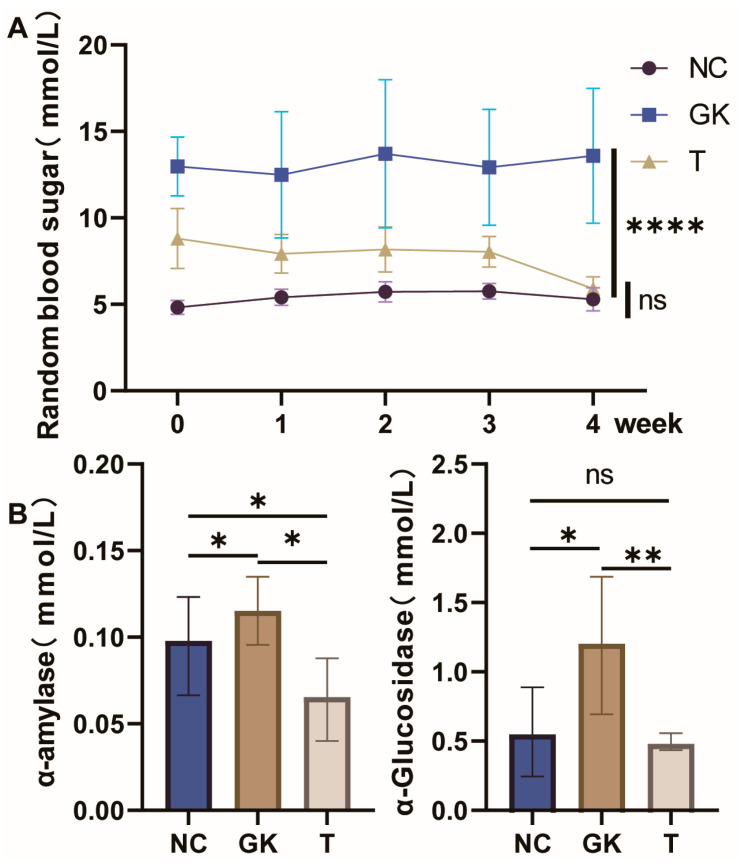
Effects of MFT on random blood glucose and intestinal α-glucosidase and α-amylase activities in hyperglycemic rats. (**A**): Random blood glucose values in mice. (**B**): α-glucosidase and α-amylase. Note: *p* < 0.05, * *p* < 0.05, ** *p* < 0.01 and **** *p* < 0.0001, ns no significant difference.

**Figure 3 foods-12-04006-f003:**
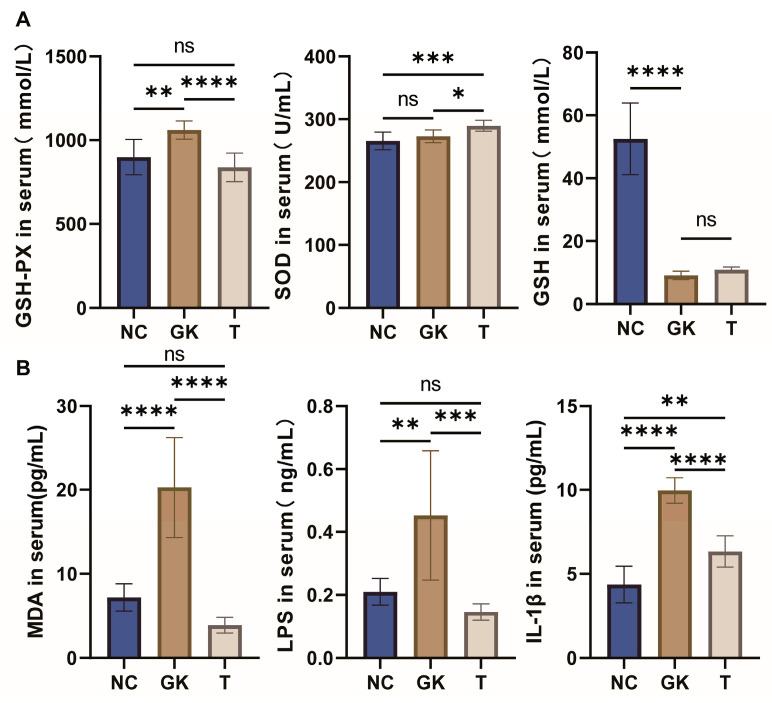
Effect of MFT on serum biochemical indicators in hyperglycemic rats. (**A**): Serum oxidative stress index. (**B**): Serum MDA, LPS, and IL-1β content. Note: *p* < 0.05, * *p* < 0.05, ** *p* < 0.01, *** *p* < 0.001, and **** *p* < 0.0001,. ns no significant difference.

**Figure 4 foods-12-04006-f004:**
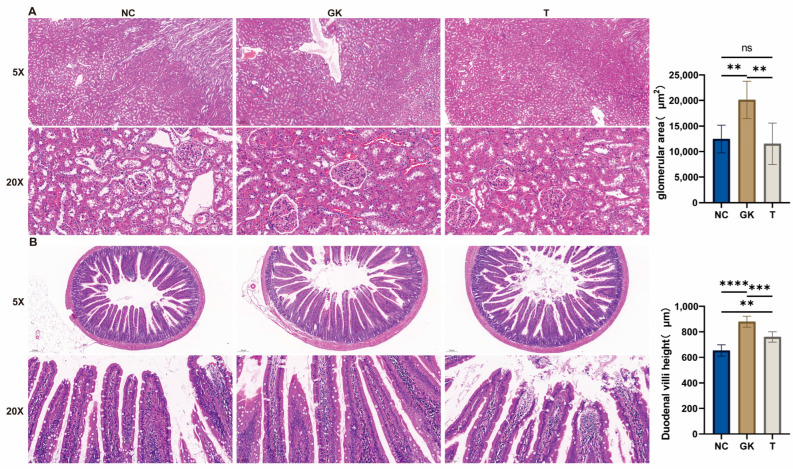
Effects of MFT on intestinal and renal structures in hyperglycemic rats. (**A**): Liver H&E section. (**B**): H&E section of duodenum. Note: *p* < 0.05, ** *p* < 0.01, *** *p* < 0.001, and **** *p* < 0.0001, ns no significant difference.

**Figure 5 foods-12-04006-f005:**
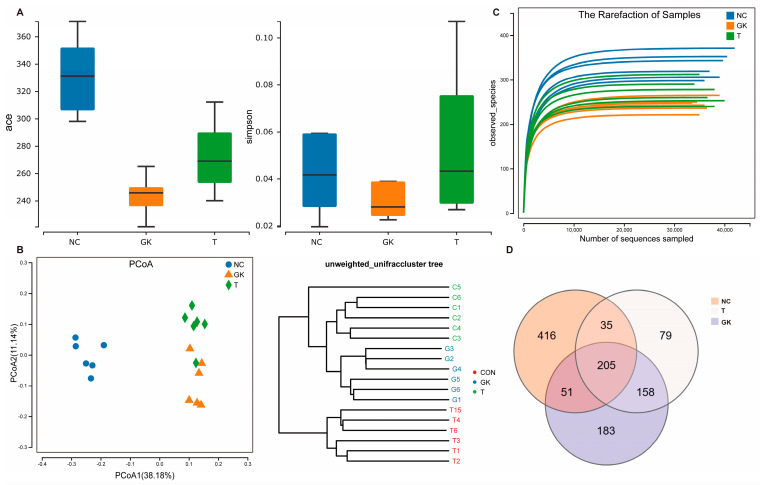
Effect of MFT on intestinal flora in hyperglycemic rats. (**A**) Venn diagram of bacterial flora. (**B**) Bacteria out. (**C**) Bacterial alpha diversity analysis. (**D**) Flora beta diversity analysis.

**Figure 6 foods-12-04006-f006:**
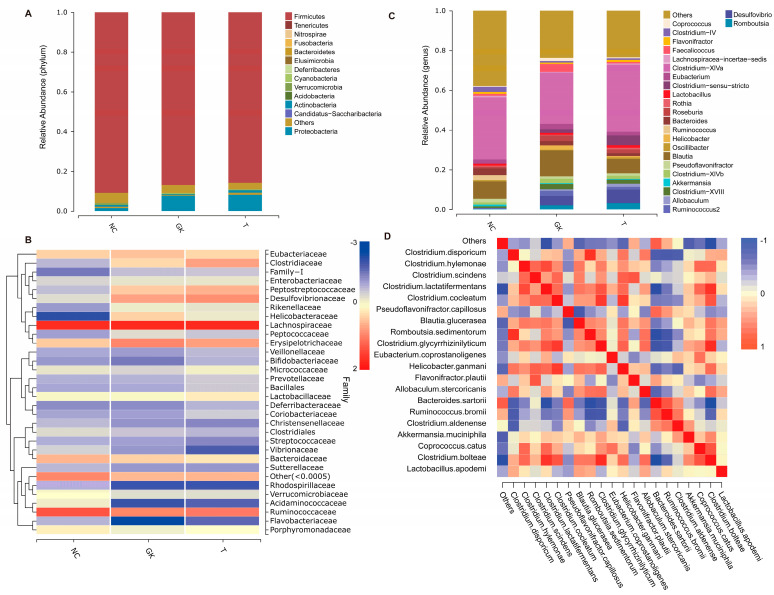
Different levels of gut flora abundance in diabetic rats. (**A**): Phylum level. (**B**): Genus level. (**C**): Family level. (**D**): Correlation of major bacterial groups at the genus level.

**Figure 7 foods-12-04006-f007:**
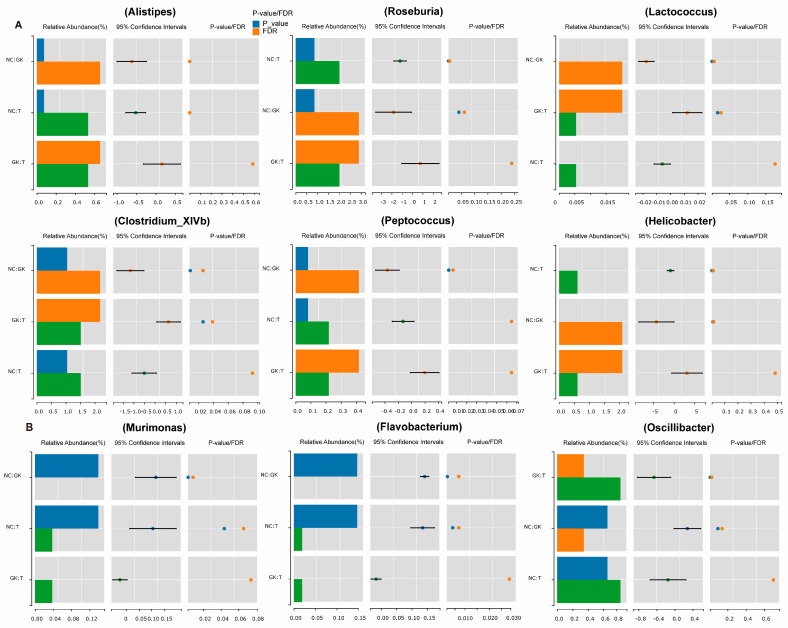
Analysis of differential bacterial groups at genus. (**A**): MFT up-regulating bacteria. (**B**): MFT down-regulating bacteria.

**Figure 8 foods-12-04006-f008:**
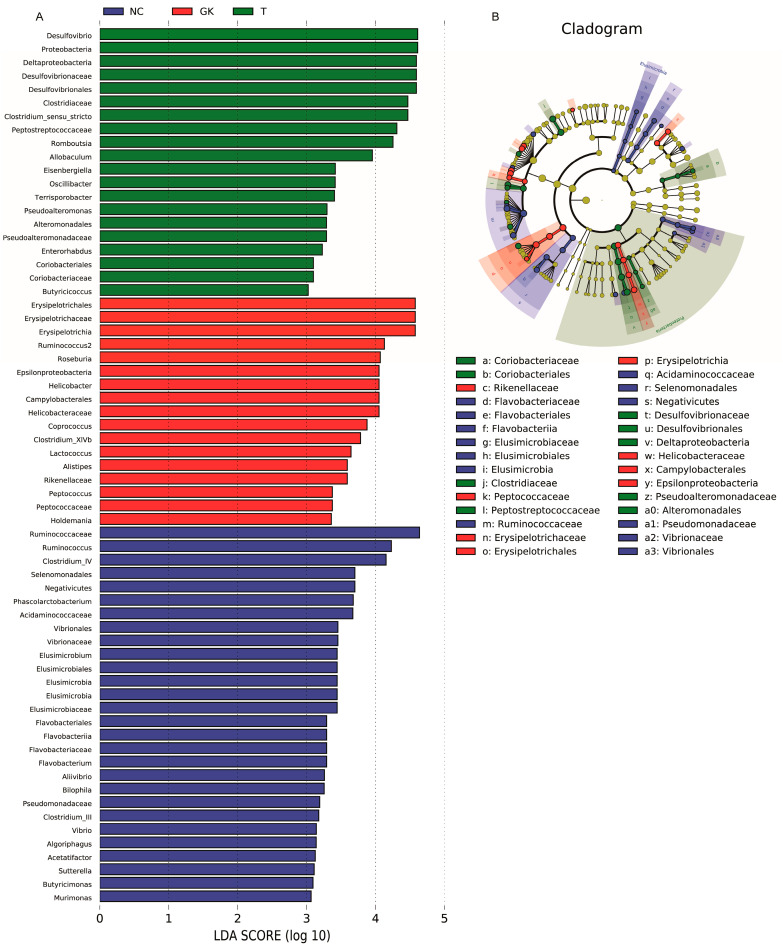
Linear Discriminant Analysis Effect Size (LEfSe) of gut microbiota in diabetic rats. (**A**): Linear discriminant analysis score of gut microbiota in diabetic rats. (**B**): LEfSe.cladogram of gut microbiota in diabetic rats.

**Figure 9 foods-12-04006-f009:**
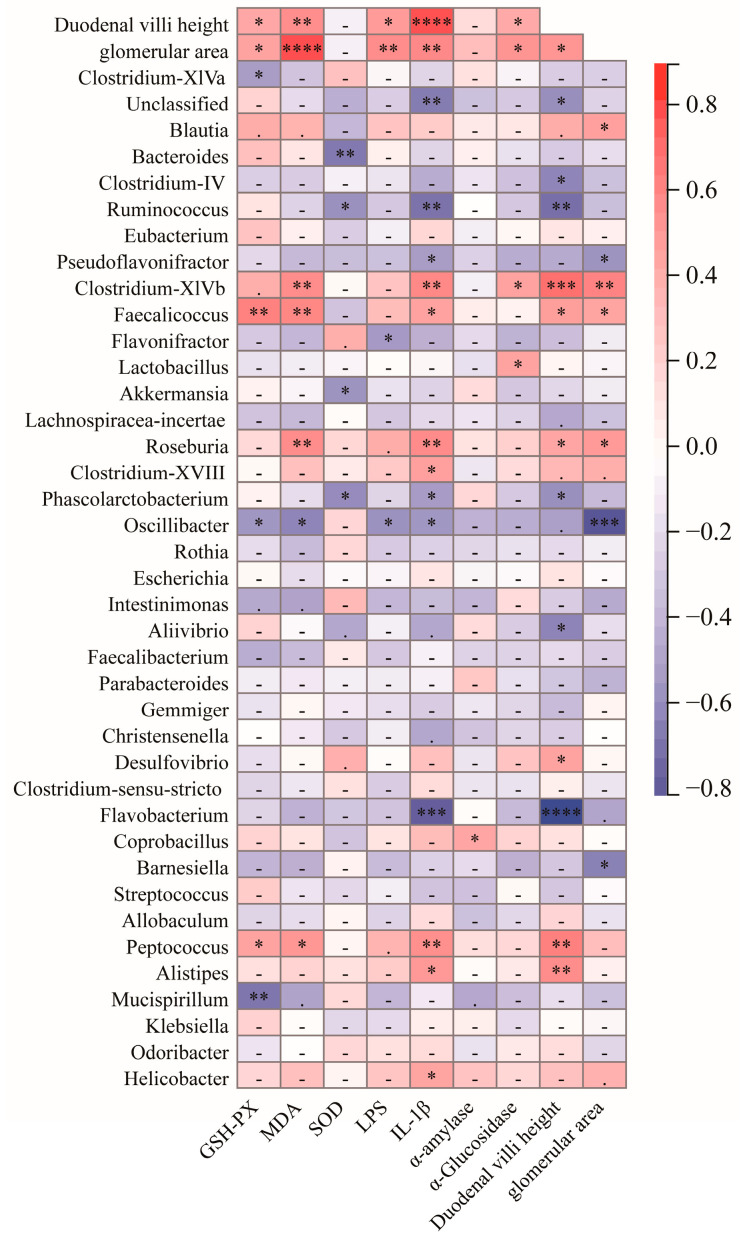
Correlation analysis between microorganisms at genus level and diabetes indicators. Note: *p* < 0.05, * *p* < 0.05, ** *p* < 0.01, *** *p* < 0.001, and **** *p* < 0.0001.

## Data Availability

All the data in this study are real and reliable, and data is contained within the Appendix A.

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
