# Peer review of "Effects of Mulberry Leaf Fu Tea on the Intestines and Intestinal Flora of Goto-Kakizaki Type 2 Diabetic Rats"

_foods, 2023, doi:10.3390/foods12214006_

Round 1
Reviewer 1 Report
Comments and Suggestions for Authors
This article is well written on the (Study on the Effects of Mulberry Leaf Fu Tea on the Intestine 2 and Intestinal Flora of Type 2 Diabetic Rats), however here are a few suggestions that will improve the quality of the manuscript if followed by the authors
1. Needs improvement in the abstract, more focused on background, aim and methodology, and conclusion part of the abstract.
2. Add numerical values and original findings of the results at least add those which are important for this study in the abstract.
3. Mention novelty of this study in the abstract
4. Explain the statistical analysis in one line in the abstract.
5. Keywords should be written in alphabetical order
6. Keep the introduction with recent supportive findings, if you can find some relevant intro of the main title recently published in 2022-2023, that could be much better
7. In the introduction, the mechanism should be elaborated on how Mulberry Leaf Fu Tea improves inflammation in the Intestine and works on the Intestinal Flora of Type 2 Diabetic Rats.
8. What is the aim of the manuscript? What new will it present? Will it attempt to answer any outstanding questions? If so, which ones? Can you write a paragraph at the end of introduction to grab the reader’s attention?
9. Why do authors choose to work only with diabetic rats? Should Mention in aims
10. Line 108 “Preparation and dosage selection of mulberry leaf and fu tea extract” The authors did not cite any reference for procedure as evidence. Also mention the equation or sample calculation method.
11. How was the search for relevant references conducted and by whom? What were the criteria for including or excluding relevant scientific work referred in this study?
12. What was the inclusion and exclusion criteria of the study subjects (rats) pre and post experiment unusual changes if any.
13. Have you conduct any safety measures, either the dose is affecting the other organs of the rats or not?
14. Give reference in section 2.4 Detection of Compounds in MFT.
15. It would be much better to add citations in all the methods to improve the quality of article.
16. Authors should mention the composition of control diet and diet along with the treatments.
17. Line 117 “Analysis of chemical components of MFT” The authors did not cite any reference for procedure as evidence.
18. Line 134 “Detection of Compounds in MFT” again The authors did not cite any reference for procedure as evidence.
19. Line 149 “Animal experiments” The authors should cite any reference for procedure as evidence.
20. Line 149 “Rat tissue morphological observation” The authors should cite any reference for procedure as evidence.
21. Line 182 “16s RNA analysis of rat intestinal flora” The authors should cite any reference for procedure as evidence.
22. For each reference you use, try to weigh the validity of the study results - how many people was it based on, and what methodology did it use? Try to confine yourself to experimental studies. If you wish you can include, perhaps separately, reviews and opinion papers.
23. Figures quality is poor. Authors should add figures which are clear and more visible to readers.
24. Please mention the full form at first and then abbreviations in the following text of the manuscript.
25. Authors should mention the reasoning in discussion along with proper comparison and justification.
26. Do your findings suggest recommendations for clinicians? I see, you suggested the Effects of Mulberry Leaf Fu Tea as a novel option for clinical care of Type 2 Diabetic patients, are you planning to explore it’s in-depth mechanisms? Can you suggest any best formulation for clinical use or this can be explored in your upcoming studies? Can you give suggestions to other researchers, who are conducting the research in the same domain as of yours?
Comments on the Quality of English LanguageExtensive editing of English language required
Author Response
This article is well written on the (Study on the Effects of Mulberry Leaf Fu Tea on the Intestine 2 and Intestinal Flora of Type 2 Diabetic Rats), however here are a few suggestions that will improve the quality of the manuscript if followed by the authors
- Needs improvement in the abstract, more focused on background, aim and methodology, and conclusion part of the abstract.
Answer: Thanks for the review. Abstract has been revised as requested.
- Add numerical values and original findings of the results at least add those which are important for this study in the abstract.
Answer: Thanks for the review. Abstract has been revised as requested.
- Mention novelty of this study in the abstract
Answer: Thanks for the review. Abstract has been revised as requested.
- Explain the statistical analysis in one line in the abstract.
Answer: Thanks for the review. Abstract has been revised as requested.
- Keywords should be written in alphabetical order
Answer: Thanks for the review. Changes have been made as requested.
- Keep the introduction with recent supportive findings, if you can find some relevant intro of the main title recently published in 2022-2023, that could be much better
Answer: Thanks for the review. It has been revised to add the relevant introduction as requested.
- In the introduction, the mechanism should be elaborated on how Mulberry Leaf Fu Tea improves inflammation in the Intestine and works on the Intestinal Flora of Type 2 Diabetic Rats.
Answer: Thanks for the review. It has been revised to add the relevant introduction as requested.
- What is the aim of the manuscript? What new will it present? Will it attempt to answer any outstanding questions? If so, which ones? Can you write a paragraph at the end of introduction to grab the reader’s attention?
Answer: Thanks for the review. Changes have been made as required.
- Why do authors choose to work only with diabetic rats? Should Mention in aims
Answer: Thanks for the review. Changes have been made as required.
- Line 108 “Preparation and dosage selection of mulberry leaf and fu tea extract” The authors did not cite any reference for procedure as evidence. Also mention the equation or sample calculation method.
Answer: Thanks for the review. Changes have been made as required.
- How was the search for relevant references conducted and by whom? What were the criteria for including or excluding relevant scientific work referred in this study?
Answer: Thanks for reviewing. Added. Relevant literature search was done by the first author through web of science, Google Scholar. Typical literature of recent years was selected for citation. Papers on hypoglycaemia of tea, mulberry leaves and other related plants were selected for citation.
- What was the inclusion and exclusion criteria of the study subjects (rats) pre and post experiment unusual changes if any.
Answer: Thanks for the review. Whether rats were modelled successfully or not was determined based on the random blood glucose values of rat rats prior to the experiment, and rats that were not modelled successfully were excluded. We set random blood glucose ≥11.1 mmol/L to be considered successful modelling.
- Have you conduct any safety measures, either the dose is affecting the other organs of the rats or not?
Answer: Thanks for the review. We observed the living condition of each group of rats in each group, and the rats in each group were in good condition, and there were no dead rats. Meanwhile, we observed the kidney morphology at the end of the experiment and found that the administration of MFT did not affect the rat's kidneys. It indicates that the dose of MFT we chose is safe.
- Give reference in section 2.4 Detection of Compounds in MFT.
Answer: Thanks for the review. Relevant references have been added.
- It would be much better to add citations in all the methods to improve the quality of article.
Answer: Thanks for the review. Relevant references have been added.
- Authors should mention the composition of control diet and diet along with the treatments.
Answer: Thanks for the review. Relevant references have been added.
- Line 117 “Analysis of chemical components of MFT” The authors did not cite any reference for procedure as evidence.
Answer: Thanks for the review. Relevant references have been added.
- Line 134 “Detection of Compounds in MFT” again The authors did not cite any reference for procedure as evidence.
Answer: Thanks for the review. Relevant references have been added.
- Line 149 “Animal experiments” The authors should cite any reference for procedure as evidence.
Answer: Thanks for the review. Relevant references have been added.
- Line 149 “Rat tissue morphological observation” The authors should cite any reference for procedure as evidence.
Answer: Thanks for the review. Relevant references have been added.
- Line 182 “16s RNA analysis of rat intestinal flora” The authors should cite any reference for procedure as evidence.
Answer: Thanks for the review. Relevant references have been added.
- For each reference you use, try to weigh the validity of the study results - how many people was it based on, and what methodology did it use? Try to confine yourself to experimental studies. If you wish you can include, perhaps separately, reviews and opinion papers.
Answer: Thanks for the review. Relevant references have been added.
- Figures quality is poor. Authors should add figures which are clear and more visible to readers.
Answer: Thanks for the review. The relevant figure has been revised.
- Please mention the full form at first and then abbreviations in the following text of the manuscript.
Answer: Thanks for reviewing. The full form has been revised.
- Authors should mention the reasoning in discussion along with proper comparison and justification.
Answer: Thanks for reviewing. Changes have been made.
- Do your findings suggest recommendations for clinicians? I see, you suggested the Effects of Mulberry Leaf Fu Tea as a novel option for clinical care of Type 2 Diabetic patients, are you planning to explore it’s in-depth mechanisms? Can you suggest any best formulation for clinical use or this can be explored in your upcoming studies? Can you give suggestions to other researchers, who are conducting the research in the same domain as of yours?
Answer: Thanks for the review. This study concludes that Mulberry Leaf Poria tea can be used as a dietary additive supplement to prevent the onset of diabetes mellitus. With emphasis on prevention rather than cure, it can be suggested that people who are prone to diabetes should consume Mulberry Leaf Poria tea on a daily basis along with moderate exercise in order to reduce body weight and control their blood glucose. Next, we plan to conduct an in-depth investigation of its mechanisms such as liver metabolism, insulin secretion and other mechanisms. Although catechins and other tea polyphenols in tea have some potential benefits, they cannot completely replace existing diabetes treatments. The effects of tea may vary from individual to individual, and factors such as dosage, type, and steeping time may also affect its effectiveness. Therefore, if you are a diabetic, it is best to consult your doctor before consuming tea or adopting any new treatment to ensure its safety and efficacy, as well as to ensure that it is coordinated with your existing treatment plan. In addition, tea should be consumed in moderation, and excessive consumption may lead to adverse reactions.
Reviewer 2 Report
Comments and Suggestions for Authors
1) How many chemicals reported by the authors? Categorize them according to their chemical class.
2) N,N-dimethylformamide: methanol: glacial 129 acetic acid = 39.5:2:1.5. Is it optimized solvent system for the separation process?
3) Give the chemical structure of theaflavins (TFs), 122 thearubigins (TRs), and theabrownin (TB), EGGC (gallatechin gallate), EGC (epigallocatechin), ECG (epicatechin gallate), EC (epicatechin), GCG (epigallocatechin gallate), DL-C (catechin), and alkaloids (caffeine, theobromine and theophylline)
4) Is there any significant relation of α-amylase and α-glucosidase lipopolysaccharide (LPS), superoxide dismutase (SOD), and glutathion peroxidase (GSH-px), glutathione (GSH), malondialdehyde (MDA) levels.
5) We detected the sequence of 320 the V3-V4 hypervariable region of the 16S rRNA of the intestinal flora of rats treated for 4 weeks, thereby analyzing the effect of MFT extract on the intestinal flora of diabetic rats. Substantiate it.
Comments on the Quality of English Language
English language and Grammatical corrections are required
Author Response
1. How many chemicals reported by the authors? Categorize them according to their chemical class.
Answer: Thanks for the review. We identified 376 compounds divided into 11 classes including catechins and their derivatives, alkaloids, flavonoids, flavonoid glycosides, phenolic acids and their derivatives, amino acids and their derivatives, organic acids, fatty acids, sugars and glycosides classes, purine bases and other.Detailed results have been added in the paper. The results of compounds obtained were also uploaded on S- table1.
2. N,N-dimethylformamide: methanol: glacial 129 acetic acid = 39.5:2:1.5. Is it optimized solvent system for the separation process?
Answer: Thanks for the review. The method is an optimised solvent system from our laboratory. Previous literature has been cited.
3. Give the chemical structure of theaflavins (TFs), 122 thearubigins (TRs), and theabrownin (TB), EGGC (gallatechin gallate), EGC (epigallocatechin), ECG (epicatechin gallate), EC (epicatechin), GCG (epigallocatechin gallate), DL-C (catechin), and alkaloids (caffeine, theobromine and theophylline)
Answer: Thanks for the review, the chemical structures of the above substances have been provided. Theobromine and theobromine are in the same class of substances, so we give the general formula. The chemical structures of all substances are shown in S-figure 1.
4. Is there any significant relation of α-amylase and α-glucosidase lipopolysaccharide (LPS), superoxide dismutase (SOD), and glutathion peroxidase (GSH-px), glutathione (GSH), malondialdehyde (MDA) levels.
Answer: Thanks for the review. Significance exists for some of the indicators. However, there are some indicators that are not significant, which we have marked in the figure, and the reason for the non-significance is that some indicators fluctuated too much in the GK rats, resulting in non-significance. In future studies we will increase the number of rats in each group to reduce the impact of chance on the experimental results.
5. We detected the sequence of 320 the V3-V4 hypervariable region of the 16S rRNA of the intestinal flora of rats treated for 4 weeks, thereby analyzing the effect of MFT extract on the intestinal flora of diabetic rats. Substantiate it.
Answer: Thanks for the review. It was determined that we were using 16s rna sequencing to detect the v3-v4 interval of gut flora in diabetic rats.